# Nucleoside Analogs That Inhibit SARS-CoV-2 Replication by Blocking Interaction of Virus Polymerase with RNA

**DOI:** 10.3390/ijms24043361

**Published:** 2023-02-08

**Authors:** Elena Matyugina, Ivan Petushkov, Sergei Surzhikov, Vasily Kezin, Anna Maslova, Olga Ivanova, Olga Smirnova, Ilya Kirillov, Irina Fedyakina, Andrey Kulbachinskiy, Sergey Kochetkov, Anastasia Khandazhinskaya

**Affiliations:** 1Engelhardt Institute of Molecular Biology, Russian Academy of Sciences, 119991 Moscow, Russia; 2Institute of Molecular Genetics, National Research Center “Kurchatov Institute”, 123182 Moscow, Russia; 3Gamaleya National Research Center for Epidemiology and Microbiology, Russian Ministry of Health, 123098 Moscow, Russia; 4Institute of Gene Biology, Russian Academy of Sciences, 119334 Moscow, Russia

**Keywords:** antiviral activity, carbocyclic and acyclic analogs of nucleosides, 6-substituted derivatives of 3H-pyrrolo [2,3-d]-pyrimidine-2-one, inhibitors, SARS-CoV-2 RNA-dependent RNA polymerase, influenza virus

## Abstract

The SARS-CoV-2 betacoronavirus pandemic has claimed more than 6.5 million lives and, despite the development and use of COVID-19 vaccines, remains a major global public health problem. The development of specific drugs for the treatment of this disease remains a very urgent task. In the context of a repurposing strategy, we previously screened a library of nucleoside analogs showing different types of biological activity against the SARS-CoV-2 virus. The screening revealed compounds capable of inhibiting the reproduction of SARS-CoV-2 with EC_50_ values in the range of 20–50 µM. Here we present the design and synthesis of various analogs of the leader compounds, the evaluation of their cytotoxicity and antiviral activity against SARS-CoV-2 in cell cultures, as well as experimental data on RNA-dependent RNA polymerase inhibition. Several compounds have been shown to prevent the interaction between the SARS-CoV-2 RNA-dependent RNA polymerase and the RNA substrate, likely inhibiting virus replication. Three of the synthesized compounds have also been shown to inhibit influenza virus. The structures of these compounds can be used for further optimization in order to develop an antiviral drug.

## 1. Introduction

Severe acute respiratory-syndrome-associated coronavirus 2 (SARS-CoV-2), first registered in Wuhan at the very end of 2019 [1], has rapidly spread all over the world, affecting >200 countries and territories. As early as on 11 March 2020, the World Health Organization (WHO) declared a pandemic due to very fast spread and morbidity rate of the infection [2]. By the end of November 2022, SARS-CoV-2 had infected more than 0.6 billion people worldwide and caused death of >6.62 mln of them [3]. A complex of pathologies in patients with this infection are referred to as coronavirus disease 2019 (COVID-19). SARS-CoV-2 targets mainly lungs and other segments of the respiratory tract inducing acute pneumonia [1,4]. However, in contrast to many other respiratory viruses, this coronaviral infection often induces severe inflammation, i.e., a cytokine storm, that in combination with hypercoagulation may lead to the development of acute respiratory distress syndrome (ARDS) [5]. In addition, COVID-19 patients often exhibit signs of extrarespiratory pathologies including gastrointestinal tract disorders, liver and pancreas dysfunction, neurological and psychiatry symptoms, and signs of heart pathology [2,6,7,8,9].

Intensive research has led to introduction to clinical practice of vaccines of various types [10]. Generally, they reduce risks of infection and development of severe COVID-19, albeit the efficacy depends on their type. Moreover, genetic diversity of the virus led to emergence of its novel variants that can bypass the pre-existing immune response, formed from vaccination or previous infection [11]. Treatment of COVID-19 is based mainly on the use of anti-inflammatory agents (glucocorticoids, monoclonal antibodies to inflammatory cytokines and their receptors, and inhibitors of respective signaling pathways) and anticoagulatory drugs [12,13,14,15,16,17]. However, SARS-CoV-2 infection still causes >1000 deaths each day, even between separate “waves” [3]. Therefore, the development of new strategies for the prevention and treatment of coronavirus infection is highly warranted.

The history of research on other viral infections has shown that the use of direct-acting antivirals (DAAs) is one of the most efficient approaches to treat infectious diseases that cannot be prevented by vaccination. Highly active antiretroviral therapy has been shown to suppress the replication of human immunodeficiency virus to undetectable levels and to prolong the life of patients with AIDS for decades [18]. The discovery of molecules that inhibit the replication of the hepatitis C virus (HCV) enables clearing the level of viral RNA from every patient with chronic hepatitis C [19]. One of the benefits of nucleoside DAAs is that they target DNA/RNA polymerases that have high genetic barriers to mutations, as such mutations can lead to the block of virus replication (for example, [20]). Therefore, it is not surprising that the development of DAAs towards SARS-CoV-2 became one of the main goals of research immediately after the identification of the pathogen. To date, three molecules have been approved for the treatment of COVID-19: paxlovid, remdesivir, and molnupiravir [21,22,23]. Paxlovid is a combination of a protease inhibitor nirmatrelvir with the antiretroviral agent ritonavir that affects the metabolism of the former molecule [24]. Remdesivir is a prodrug of a nucleoside inhibitor of viral RNA-dependent RNA polymerase (RdRp) [25]. Molnupiravir is a depot form of the modified nucleoside N^4^-hydroxycytidine that is known to induce lethal mutagenesis during the replication of RNA viruses and, according to Zhou et al., during replication of the host cell genome [26]. Although the efficacy of remdesivir remains uncertain [27], the other two drugs show clinical efficacy including in the era of Omicron variants of SARS-CoV-2 [28]. Therefore, additional endeavors are needed to develop effective DAAs.

In 2020–2021, the collection of analogs of heterocyclic bases and nucleosides synthesized in the Engelhardt Institute of Molecular Biology RAS was tested on the ability to inhibit Vero cell death induced by SARS-CoV-2 virus infection with the PIK35 strain [29]. In total, more than 200 compounds have been evaluated, 2 of which, 1-(4’-hydroxy-2’-cyclopenten-1-yl)-6-(4-pentylphenyl)-3H-pyrrolo [2,3-d]-pyrimidine-2-one (compound **1**) and 1-(4’-hydroxy-2’-cyclopenten-1-yl)-6-(4-tert-butylphenyl)-3H-pyrrolo [2,3-d]-pyrimidine-2-one (compound **2**), showed dose-dependent ability to inhibit the reproduction of SARS-CoV-2 with EC_50_ values of 21 and 53 μM [29]. In order to identify the structural determinants of antiviral activity, we carried out clarifying studies of the activity of these compounds in various test systems and synthesized different types of analogs of compounds **1** and **2** to investigate the dependence of their activity on the structure. As directions for the modification of compound **1**, we chose the replacement of the NH group in the five-membered base ring with oxygen, resulting in 3H-furano [2,3-d]-pyrimidine analogs **3**–**5**, and the replacement of the carbocyclic fragment with a ribose one (compounds **4** and **6**) or acyclic (compounds **5** and **7**) residue (Figure 1).

Analogs of compound **2**, 1- and 1,3-bis carbocyclic derivatives of 5-(4-tert-butylphenylamino)uracil **8** and **9** and 5-(4-tert-butylphenyloxy)uracil **10** and **11**, as well as the corresponding ribocompounds **12** and **13**, were synthesized (Figure 2).

We tested the antiviral activity of these compounds in cell-based assays and demonstrated that some of them have potent effects on virus replication. Furthermore, we revealed that several of the tested compounds can inhibit the interactions of SARS-CoV-2 RdRp with the RNA substrate in vitro; suggesting that their main target in vivo is the viral replication machinery.

## 2. Results

### 2.1. Chemistry

Compounds **1**–**3** were obtained as described previously [30]. Derivatives **8**–**11** were synthesized by condensation of epoxycyclopentene with 5-(4-tert-butylphenylamine)- or 5-(4-tert-butylphenyloxy)-uracil in the presence of a palladium catalyst according to the Trost method [31].

The 5-substituted analogs of ribonucleosides **12** and **13** were obtained in acceptable yields (62% and 69%) from the corresponding uracil derivatives [31] by the Vorbruggen method followed by removal of the protecting groups (Figure 1).

Compounds **17** (Figure 2) and **19** (Figure 3), which are key intermediates in the synthesis of ribo- and acyclic analogs of furano- and pyrrolopyrimidine nucleosides, respectively, were obtained by the same method starting from 5-ioduracil. The reaction of compound **17** with 1-ethynyl-4-pentylbenzene in the presence of 10% palladium on carbon and copper iodide followed by the treatment with an ammonia solution in methanol led to a mixture of furano- and pyrrolopyrimidine products **4** and **6** (Figure 2), which was separated by preparative layer chromatography, eluting with a mixture of chloroform and methanol (9:1). The yields of products **4** and **6** were 31% and 44%, respectively.

Acyclic analogs **5** and **7** were synthesized similarly starting from compound **19** (Figure 3). The yields of products **5** and **7** were 35% and 40%, respectively.

### 2.2. Biological Evaluation

The antiviral activity of all synthesized compounds towards SARS-CoV-2 was assessed by two approaches. First, we quantified changes in the amount of viral RNA during virus replication in the conditioned medium (to measure the reduction in virion production) in the presence of each compound taken at a fixed concentration (Δlog_10_(RNA)) (Table 1). Second, we analyzed changes in the infectivity of the virus, by measuring changes in the Tissue Culture Infectious Dose values in Vero E6 cells (Δlog_10_(TCID_50_)). A previously studied anti-SARS-CoV-2 agent N4-hydroxycitidine (NHC) was used as a control [26,29].

These two approaches gave complementary results, presented in Table 1. Two of the compounds (**6** and **7**) were inactive towards SARS-CoV-2 in both assays. The highest activity was displayed by the previously described compound **1**, which was active in both assays and significantly decreased the amount of viral RNA (>10-fold) and the TCID_50_ value (~10-fold)**.** Moderate activity was registered for the newly developed compounds **3**, **5**, **8**, **11**, **13**, and **14**, which could inhibit virus replication or infectivity at least 10-fold in one of the 2 assays (Table 1). Compound **8**, albeit showing visible reduction in the viral RNA titer, provided weak protection of Vero cells in the infectivity test. On the contrary, compounds **5** and **14** only weakly affected the RNA titer but had stronger effects on the TCID_50_ values. Other tested compounds showed weak antiviral activity in both assays. The toxicity of all tested compounds was measured by the standard MTT assay in Vero E6 cells (CD_50_ Vero E6, Table 1). Noteworthy, all of them were almost nontoxic at the effective concentration, as just a few of them inhibited cell growth at concentrations of 150 μM or above.

We then analyzed the activity of these compounds towards the influenza virus by measuring changes in the virus infectivity (Δlog_10_(TCID_50_)) in MDCK cells, which is a standard cell line highly permissive to this virus. Clear protection of cells against influenza-virus-induced cell death was demonstrated by several compounds at 50 μM concentration (Table 1). Furthermore, we calculated concentrations of the compounds at which they reduced virus titers by 50% (IC_50_). The highest antiviral activity was observed for compounds **2**, **5**, and **14**, which had IC_50_ values of ~1 μM, comparable with IC_50_ for oseltamivir (Table 1). The toxicity of these compounds in MDCK cells (CD_50_ MDCK, Table 1) was comparable with Vero E6 cells. The highest selectivity index (SI, the ratio of CD_50_ to IC_50_) exceeding 500 was observed for the acyclic derivative of 6-substituted 3H-pyrrolo [2,3-d]-pyrimidine-2-one **5**.

### 2.3. Inhibition of RdRp Activity In Vitro

Since all obtained compounds are nucleoside mimetics (Figure 1 and Figure 2), it was natural to propose that their target in vivo could be the replication machinery of SARS-CoV-2. To test this hypothesis, we analyzed the effects of these compounds on the activity of the SARS-CoV-2 replicase in vitro, using purified RNA-dependent RNA polymerase (RdRp) holoenzyme consisting of the catalytic subunit nsp12 and two accessory subunits, nsp7 and nsp8. The reactions were performed with compounds **1**, **3**, **5**, **8**, **11**, and **14**, which noticeably inhibited virus replication in cell-based assays, and with compounds **9**, **10**, and **12**, which showed weaker effects in cell cultures. RdRp was incubated with each of the compounds and the reaction was initiated by adding a pre-annealed primer–template RNA substrate (Figure 3, top) and nucleotides. In accordance with our previous analysis of RdRp activity with this RNA substrate [32], in the absence of inhibitors, RdRp extended the RNA primer until the end of the template strand (Figure 3, control reactions with DMSO). It was found that several of the tested compounds could inhibit RNA extension (Figure 3A,D). The strongest effects were observed for compounds **1**, **3**, **5**, **10**, and **14**, which completely blocked RNA synthesis. Overall, these data corroborate our findings from the cell infectivity models. In particular, four of these compounds (**1**, **3**, **5**, and **14**) are also among the strongest inhibitors in cell-based assays. Compound **10**, while being less efficient in cell culture, could nevertheless fully inhibit RdRp in vitro. Furthermore, compounds **9** and **12**, which are less efficient in cell-based assays, also have weaker effects on the RdRp activity in vitro (Figure 3A,D).

To test whether the same compounds can act on pre-assembled replicative complexes, we first incubated RdRp with the RNA substrate and then added the inhibitors and nucleotides. Surprisingly, these compounds could not efficiently inhibit RdRp activity under these conditions (Figure 3B,D). This suggested that the binding sites of these inhibitors on RdRp may be masked by RNA in the replicative complex. Alternatively, it could be proposed that these compounds might interact with the RNA substrate instead of RdRp and prevent its binding to RdRp. To rule out this possibility, we first preincubated the inhibitors with RNA and then added RdRp and nucleotides. No strong inhibition was observed in this case (Figure 3C,D), suggesting that the analyzed compounds act on RdRp rather than on the RNA substrate.

To elucidate the mechanism of RdRp inhibition by the tested compounds, we analyzed their effects on RNA binding by RdRp using an electrophoretic mobility shift assay (EMSA). In the absence of inhibitors, RdRp bound the radiolabeled primer–template RNA substrate with high efficiency (Figure 4A). However, RNA binding was inhibited if RdRp was first incubated with the compounds that inhibited RdRp activity in the primer extension assay (Figure 4A, left panel, and Figure 4B). In particular, compounds **1**, **10**, and **14** fully prevented RNA binding, in agreement with their strong effects on primer extension (Figure 3A). In contrast, much weaker or no effects of the same compounds on RNA binding were observed if they were added to preformed RdRp-RNA complexes (Figure 4A, right panel, and Figure 4B). Accordingly, these compounds did not inhibit primer extension in preformed RdRp-RNA complexes (Figure 3B). It can therefore be concluded that the inhibitors tested here prevent RNA binding by RdRp but cannot act on active RdRp-RNA complexes.

To determine the range of inhibitory concentrations of these compounds in vitro, we titrated RdRp with compounds **1**, **5**, and **14**, then added the RNA substrate and measured the efficiency of RNA extension (Figure 5). The resulting IC_50_ values for these compounds were 44 ± 2, 72 ± 20, and 244 ± 58 μM, respectively, which corresponded to the range of their effective concentrations in cell-based assays (Table 1).

## 3. Discussion

In this study, we obtained and characterized a series of nucleoside derivatives that can inhibit the reproduction of the SARS-CoV-2 virus in cell culture. We found that several of the newly synthesized compounds can decrease the titer of virus RNA after infection and/or decrease the virus infectivity with efficiencies that are comparable with the original compound **1** described previously [29]. Remarkably, some of the tested compounds, including compounds **5** and **14**, could also inhibit the replication of the influenza virus, suggesting that they might target a common step in the viral cell cycle and could be used for further development of broad spectrum antivirals.

We demonstrated that several of the synthesized compounds can inhibit the activity of SARS-CoV-2 RdRp in biochemical assays. In particular, compounds **1**, **3**, **5**, and **14** can inhibit both virus replication in cell culture and RdRp activity in vitro, while several compounds that are less efficient in vivo also have lower activity against RdRp in vitro. The IC_50_ values for inhibition of RdRp activity by compounds **1** and **5** (Figure 5) exactly correspond to the concentration at which they can inhibit virus replication in cell-based assays (Table 1). Although the IC_50_ value for compound **14** is somewhat higher than its active concentration in cell culture, it can also fully inhibit RdRp in vitro. This suggests that RdRp is likely the natural target of these compounds during virus replication. Some discrepancies between the inhibitory activities of the tested compounds in RdRp assays and in cell-based assays (e.g., relatively low activity of compound **10** in cell cultures despite its ability to fully inhibit RdRp in vitro) can likely be explained by different cell permeability and/or differences in the metabolism of these compounds in vivo.

As we demonstrated previously, compounds **1** and **2** (and probably their derivatives tested here) cannot be phosphorylated in vivo. This suggests that they cannot be incorporated into nascent RNA by RdRp and cannot act as chain terminators or promote lethal mutagenesis similarly to remdesivir or molnupiravir [21,22,25]. Indeed, we observed that none of the tested compounds can affect the RNA extension reaction performed by RdRp-RNA complexes but can efficiently inhibit RdRp activity if added before the RNA substrate (Figure 3), suggesting that they interfere with RNA binding rather than with RNA synthesis.

Since the tested compounds can inhibit RdRp activity by preventing its interactions with RNA, we propose that their binding sites are likely located within the RNA-binding channel of RdRp. Docking of compound **1** on the SARS-CoV-2 RdRp structure using the SwissDock web service revealed multiple potential binding sites, some of which were indeed located within the RNA-binding channel (Figure 6). However, no preferred site with the highest binding energy could be revealed in this modeling, consistent with the relatively low affinity (IC_50_ values) of the tested compounds to RdRp. Thus, nucleoside derivatives tested here might potentially target several alternative sites on the RNA-binding surface of RdRp and further experiments are required to establish their exact binding mode to the replication complex of SARS-CoV-2.

Previously, several non-nucleoside SARS-CoV-2 inhibitors were hypothesized to interfere with RdRp-RNA interactions. A pyridobenzothiazole compound, HeE1-2Tyr, initially characterized as an inhibitor of flavirus RdRp interacting with its RNA-binding sites [36], was shown to inhibit SARS-CoV-2 RdRp with IC_50_ of 27 μM [37]. Furthermore, a helquat-like compound PR673 was shown to inhibit SARS-CoV-2 RdRp with IC_50_ of ~4 μM [38]. However, the effects of both compounds on RNA binding by SARS-CoV-2 RdRp were not tested. The only compound for which a direct effect on RNA binding by SARS-CoV-2 RdRp was demonstrated in vitro is an antiparasitic drug suramin, which interacts with RdRp with micromolar affinity and interferes with template and primer RNA binding [39]. The new compounds characterized in our study provide the second example of non-nucleoside inhibitors of SARS-CoV-2 RdRp with a defined mechanism of action and may serve as a starting point for developing more efficient drugs preventing virus replication.

## 4. Materials and Methods

### 4.1. Chemistry Experimental

#### 4.1.1. General

Commercial reagents were purchased from Acros Organics, Aldrich, and Fluka. Solvents were used without further purification and distillation. Column chromatography was carried out on silica gel 60 0.040–0.063 mm (Merck, Darmstadt, Germany). Thin layer chromatography was performed on silica gel 60F_254_ aluminum foil (Merck, Darmstadt, Germany). NMR spectra were recorded on an AMX III-400 spectrometer (Bruker, Billerica, USA) with an operating frequency of 400 MHz and 300 MHz for ^1^H NMR (solvent—DMSO-d_6_, Me_4_Si as internal standard) and 100.6 MHz for ^13^C NMR. UV spectra were recorded on an Ultrospec 3100 pro spectrophotometer (Amersham Biosciences, Chicago, USAs) in ethanol. High-resolution mass spectra were recorded on a Bruker Daltonics MicrOTOF-Q II device using electrospray ionization mass spectrometry (ESI-MS). The measurements were carried out in the mode of positive ions in accordance with the previously applied conditions [32].

#### 4.1.2. Compound Synthesis and Characterization


**1-(β-D-Ribofuranosyl)-5-(4-tert-butylphenylamine)-uracil (12)**


5-(4-tert-Butylphenylamine)-uracil (1.38 mmol, 0.336 g) was silylated in 1,1,1,3,3,3-hexamethyldisilazane (50 mL) in the presence of ammonium sulfate (2 mg) for 4 h. The resulting solution was evaporated to dryness under high vacuum and then coevaporated with toluene (2 × 50 mL) and 1,2-dichloroethane (40 mL) to remove traces of 1,1,1,3,3,3-hexamethyldisilazane. The semicrystalline residue was dissolved in 1,2-dichloroethane (20 mL), β-D-1,2,3,5-tetraacetate ribose (1.106 mmol, 0.352 g) was added with stirring to one portion, and then trimethylsilyl ester of trifluoromethanesulfonic acid (1.15 mmol, 0.255 g) was added dropwise. The resulting mixture was stirred at 50 °C for 4 h. The reaction progress was monitored by TLC in the CH_2_Cl_2_/C_2_H_5_OH (20:1) system. After completion of the reaction, the reaction mixture was cooled and added dropwise to a vigorously stirred mixture of saturated sodium bicarbonate solution and methylene chloride (100 mL 1/1 by volume). The resulted mixture was stirred for 30 min. The organic layer was separated, washed with water (2 × 50 mL), dried over anhydrous sodium sulfate for 12 h, and evaporated to dryness. The residue was separated by silica gel column chromatography. The product was eluted with a gradient of ethanol in methylene chloride from 1:40 to 1:20. Fractions containing target product **14** were combined and evaporated to dryness.

To remove the protecting groups, the crystalline residue was dissolved in ethanol (10 mL) and 32% NH_4_OH (10 mL) was added and left at 20°C for 10 h. The solvents were then evaporated to dryness. The final product **12** was purified by crystallization from hot ethanol or silica gel column chromatography in a gradient of ethanol in methylene chloride from 1:20 to 1:9. The fractions containing the final product were combined and evaporated to dryness. The total yield of compound **12** was 69.7% (0.301 g). M.p. 199°C; UV: λ_max_ 263 nm, λ_min_ 241 nm; ESI-MS: C_19_H_25_N_3_O_6_ calculated for [M + H]^+^ 392.4183, found m/z 392.4185. ^1^H-NMR (DMSO-d_6_) δ, ppm: 1.24 (9H, s, 3xCH_3_), 3.63–3.53 (2H, m, H-5′_a,b_), 3.86–3.82 (1H, m, H-4′), 4.00–3.96 (1H, dt, J = 3.73, 4.92 Hz, H-3′), 4.12–4.06 (1H, dd, J = 5.75, 5.82 Hz, H-2′), 5.07–5.04 (2H, m, 5′-OH, 3′-OH), 5.36–5.34 (1H, d, J = 5.87 Hz, 2′-OH), 5.89–5.87 (1H, d, J = 5.96 Hz, H-1′), 7.18–6.76 (5H, m, NH, C_6_H_4_), 7.75 (1H, s, H-6), 11.54 (1H, s, NH); ^13^C NMR (DMSO-d_6_) δ, ppm: 31.9, 34.1, 61.6, 70.9, 73.9, 85.5, 88.0, 114.9, 118.1, 125.8, 128.0, 141.1, 143.2, 150.1, 161.8.


**1-(β-D-Ribofuranosyl)-5-(4-tert-butylphenoxy)-uracil (13)**


5-(4-tert-Butylphenoxy)-uracil (2.05 mmol, 0.535 g) was silylated as described above in 1,1,1,3,3,3-hexamethyldisilazane (50 mL) in the presence of ammonium sulfate (2 mg) for 4 h. The resulting semicrystalline residue was dissolved in acetonitrile (20 mL), 1-O-acetyl-β-D-2,3,5-tri-O-acetyl-β-D-ribofuranose (1.78 mmol, 0.901 g) was added at stirring, and then trimethylsilyl ester of trifluoromethanesulfonic acid (2.14 mmol, 0.475 g) was added dropwise. The resulting mixture was stirred at 50 °C for 4 h. The reaction progress was monitored by TLC in the CH_2_Cl_2_/C_2_H_5_OH (20:1) system. The reaction mixture was worked up and the protected nucleoside analog was isolated as described above.

The crystalline residue containing the protected nucleoside analog **15** was dissolved in ethanol (20 mL), 32% NH_4_OH (10 mL) was added and left at 20 °C for 10 h. The solvents were then evaporated to dryness. The final product **13** was purified by crystallization from hot ethanol or silica gel column chromatography with a CH_2_Cl_2_/C_2_H_5_OH gradient from (20:1) to (9:1). Fractions containing the target product **13** were combined and evaporated to dryness. The total yield of compound **13** was 62.22% (0.436 g). M.p. 215 °C; UV: λ_max_ 272,5 nm, λ_min_ 242 nm; ESI-MS: C_19_H_24_N_2_O_7_ calculated for [M + H]^+^ 392.403, found m/z 392.435; ^1^H-NMR (DMSO-d_6_) δ, ppm: 1.26 (9H, s, 3xCH_3_), 3.67–3.50 (2H, m, H-5′_a,b_), 3.87–3.84 (1H, m, H-4′), 4.00–3.36 (1H, dd, J = 4.8, 4.95 Hz, H-3′), 4.10–4.04 (1H, dd, J = 5.11, 5.15 Hz, H-2′), 5.05–5.03 (1H, d, J = 5.21 Hz, 3′-OH), 5.12–5.09 (1H, t, J = 4.79, 5′-OH), 5.41–5.39 (1H, d, J = 5.46 Hz, 2′-OH), 5.81–5.80 (1H, d, J = 4.99 Hz, H-1′), 6.90–6.85 (2H, m, C_6_H_4_), 7.33–7.28 (2H, m, C_6_H_4_), 8.15 (1H, s, H-6), 11.66 (1H, s, NH); ^13^C NMR δ, ppm: 31.8, 34.3, 60.9, 70.1, 74.3, 85.2, 88.6, 114.9, 126.6, 129.9, 132.7, 144.9, 150.4, 156.1, 159.7.


**1-(β-D-(2′,3′,5′-Triacetylribofuranosyl))-5-ioduracil (17)**


5-ioduracil (1 mmol) was silylated by refluxing in 1,1,1,3,3,3-hexamethyldisilazane (20 mL) in the presence of ammonium sulfate (1 mg) and pyridine (2 mL) for 4 h. The resulting clear solution was evaporated to dryness on a water jet pump and dried under high vacuum. The residue was dissolved in acetonitrile (15 mL) and β-D-1,2,3,5-tetraacetate ribose (1 mmol) and trimethylsilyl trifluoromethanesulfonate (1.5 mmol) were added. The resulting mixture was stirred for 18 h. The reaction progress was monitored by TLC in the CHCl_3_/C_2_H_5_OH (98:2) system. The product was purified by silica gel column chromatography. The product was eluted with a gradient of ethanol in chloroform from (1:99) to (2:98). The product **17** was obtained in the form of a yellowish powder, the yield was 87% (0.470 mg). ^1^H-NMR (CDCl_3_) δ, ppm: 2.10 (3H, s, CH_3_), 2.17 (3H, s, CH_3_), 2.26 (3H, s, CH_3_), 4.35–4.44 (3H, m, H-2′, H-3′, H-4′), 5.34–5.36 (2H, m, H-5′a,b), 6.08–6.10 (1H, m, H-1′), 7.91 (1H, s, H-6), 8.76 (1H, s, NH). ^13^C NMR (CDCl_3_) δ, ppm: 20.4, 20.5, 21.1, 63.0, 69.5, 70.2, 73.1, 77.2, 80.4, 87.2, 143.7, 149.8, 159.4, 169.6, 170.1.


**3-(β-D-(2′,3′,5′-Triacetylribofuranosyl))-6-(4-pentylphenyl)-3H-furano [2,3-d]-pyrimidine-2-one (18)**


To the solution of 1-(β-D-(2′,3′,5′-triacetylribofuranosyl))-5-iodouracil 17 (230 mg, 0.46 mmol) in acethonitrile (10 mL) CuI (11 mg, 0.1 mmol), 10% Pd/C (50 mg), Et_3_N (465 mg, 4.6 mmol), and 1-ethynyl-4-pentylbenzene (103 mg, 116 μL, 0.6 mmol) were added and the reaction mixture was refluxed for 4 h. The progress of the reaction was monitored by TLC in the CHCl_3_:CH_3_OH (98:2) system. Solvents were evaporated to dryness in vacuo, and the residues were purified using column chromatography on silica gel in the CHCl_3_:CH_3_OH (98:2) system and re-purified in the Hexane:EtOAc (3:2) system. The yield of product **18** as a white powder was 140 mg (56%). ^1^H-NMR (CDCl_3_) δ, ppm: 0.86–0.96 (3H, m, CH_3_), 1.31–1.38 (4H, m, 2xCH_2_), 1.63–1.68 (2H m, CH_2_), 2.12 (3H, s, CH_3_), 2.15 (3H, s, CH_3_), 2.20 (3H s, CH_3_), 2.64–2.69 (2H, m, CH_2_), 4.45–4.51 (3H, m, H-4′, CH_2_), 5.36 (1H, t, J = 5.6 Hz, H-3′), 5.48 (1H, dd, J = 5.5, 3.8 Hz, H-2′), 6.29 (1H, d, J = 3.8 Hz, H-1′), 6.65 (1H, s, H-5), 7.27–7.30 (2H, m, Ph), 7.70 (2H, d, J = 8.2 Hz, Ph), 8.21 (1H, s, H-4). ^13^C NMR (CDCl_3_) δ, ppm: 14.0, 20.5, 20.5, 20.9, 22.5, 30.9, 31.4, 35.9, 62.7, 69.5, 74.2, 79.8, 90.0, 96.3, 109.1, 125.1, 125.6, 129.1, 134.5, 145.5, 156.9, 169.5, 169.6, 170.1.


**3-(β-D-Ribofuranosyl)-6-(4-pentylphenyl)-3H-furano [2,3-d]-pyrimidine-2-on (4) and 3-(β-D-ribofuranosyl)-6-(4-pentylphenyl)-3H-pyrrolo [2,3-d]-pyrimidine-2-on (6)**


Compound **18** (120 mg, 0.22 mmol) was dissolved in 32% NH_3_ in methanol (15 mL). The reaction mixture was kept at 36°C for 72 h. Solvent then was evaporated and a new portion of 32% NH_3_ in methanol was added (15 mL). The procedure was repeated until it reached approximately 50% conversion of compound **4** to compound **6** controlling the progress of the reaction by TLC. The solvent then was evaporated and the products were purified using preparative chromatography in the CHCl_3_:CH_3_OH (9:1) system.

Yield of **4** as a pale yellow powder was 28 mg (31%). ^1^H-NMR (DMSO-d_6_) δ, ppm: 0.83–0.91 (3H, m, CH_3_), 1.26–1.33 (4H, m, 2xCH_2_), 1.57–1.62 (2H, m, CH_2_), 2.62 (2H, t, J = 7.6 Hz, CH_2_), 3.65–3.89 (2H, m, CH_2_), 3.97–4.06 (3H, m, 3xOH,), 5.02–5.03 (1H, m, H-4′), 5.32 (1H, t, J = 5.0 Hz, H-3′), 5.58–5.60 (1H m, H-2′), 5.89 (1H, d, J = 1.8 Hz, H-1′), 7.21 (1H, s, H-5), 7.33 (2H, d, J = 8.3 Hz, Ph), 7.75 (2H, d, J = 8.3 Hz, Ph), 8.97 (1H, s, H-4). ^13^C NMR (CDCl_3_) δ, ppm: 14.4, 22.4, 29.4, 30.9, 31.3, 35.4, 59.9, 68.5, 75.3, 84.6, 92.1, 99.1, 107.5, 125.1, 126.4, 129.5, 138.7, 144.6, 154.5, 171.6. HRMS m/z: calculated for C_22_H_26_N_2_O_6_ [M+H]+ 415.1864; found [M+H]^+^ 415.1861.

Yield of **6** as white powder was 40 mg (44 %). ^1^H-NMR (CD_3_OD) δ, ppm: 0.79–0.83 (3H, m, CH_3_), 1.24–1.27 (4H, m, 2xCH_2_), 1.52–1.57 (2H, m, CH_2_), 2.62 (2H, dd, J = 8.5, 6.8 Hz, CH_2_), 3.84 (2H, ddd, J = 52.6, 12.4, 2.6 Hz, CH_2_), 3.95–4.12 (3H, m, H-2′, H-3′, H-4′), 5.95 (1H, d, J = 1.9 Hz, H-1′), 6.51 (1H, s, H-5), 7.15–7.18 (2H, m, Ph), 7.53–7.56 (2H, m, Ph), 8.81 (1H, s, H-4). ^13^C NMR (CD_3_OD) δ, ppm: 12.9, 22.1, 29.3, 30.8, 31.2, 35.2, 60.0, 68.6, 75.7, 84.5, 92.7, 96.0, 111.0, 125.0, 128.0, 128.7, 136.3, 143.8, 155.9, 159.5. HRMS m/z: calculated for C_22_H_27_N_3_O_5_ [M+H]^+^ 414.2023, [M+Na]+ 436.1843; found [M+H]+ 414.2022, [M+Na]+ 436.1845.


**1-((2-(Acetoxy)ethoxy)methyl)-5-iodouracil (19)**


5-Ioduracil (238 mg, 1 mmol) was refluxing in 1,1,1,3,3,3-hexamethyldisilazane (20 mL) with ammonium sulfate (1 mg) and pyridine (2 mL) during 4 h. Then, the reaction mixture was evaporated to dryness and dried under high vacuum. The residue was dissolved in acetonitrile (15 mL) followed by the addition of trimethylsilyl trifluoromethanesulfonate (1.5 mmol) and (2-acetoxyetoxy)methyl acetate (1 mmol). The reaction mixture was stirred for 18 h. The progress of the reaction was monitored by TLC. The product was purified by column chromatography on silica gel eluting with CHCl_3_/C_2_H_5_OH (98:2). The product **19** was obtained as a yellowish powder with 87% yield. ^1^H NMR (Methanol-*d*_4_) δ, ppm: 2.06 (3H, s, CH_3_), 3.85–3.71 (2H, m, CH_2_), 4.25–4.15 (2H, m, CH_2_), 5.17 (2H, s, CH_2_), 7.86 (1H, s, H-6). ^13^C NMR (75 MHz, Methanol-*d*_4_) δ, ppm: 20.5, 63.0, 67.7, 68.9, 71.1, 147.9, 148.3, 151.2, 161.2.


**3-((2-(Acetoxy)ethoxy)methyl)-6-(4-pentylphenyl)-3H-furano [2,3-d]-pyrimidine-2-one (20a)**


To the solution of 1-((2-(acetoxy)ethoxy)methyl)-5-iodouracil **19** (0.48 mmol, 170 mg) in acethonitrile (10 mL) CuI (0.1 mmol, 18 mg), 10% Pd/C (0.04 mmol, 45 mg), Et_3_N (2.9 mmol, 290 mg), and 1-ethynyl-4-pentylbenzene (0.72 mmol, 124 mg) were added and the reaction mixture was refluxed for 4 h. The progress of the reaction was monitored by TLC in the CHCl_3_:CH_3_OH (98:2) system. The solvents were evaporated to dryness in vacuo, and the residues were purified using column chromatography on silica gel in the Hexane:EtOAc (3:2) system.

The yield of **20a** as a white powder was 117 mg (61%). ^1^H-NMR (CDCl_3_) δ, ppm: 0.87–0.95 (3H, m, CH_3_), 1.35 (4H, dq, J = 7.9, 4.2 Hz, 2xCH_2_), 1.66 (2H, q, J = 7.6 Hz, CH_2_), 2.07 (3H, s, CH_3_), 2.63–2.68 (2H, m, CH_2_), 3.91–3.94 (2H, m, CH_2_), 4.23–4.26 (2H, m, CH_2_), 5.52 (2H, s, CH_2_), 6.69 (1H, s, H-5), 7.26–7.29 (2H, m, Ph), 7.68–7.71 (2H, m, Ph), 8.04 (1H, s, H-4). ^13^C NMR (CDCl_3_) δ, ppm: 14.0, 20.8, 22.5, 30.9, 31.4, 35.9, 63.1, 68.5, 79.5, 96.3, 109.4, 125.1, 125.6, 129.1, 137.9, 145.5, 155.7, 156.9, 170.8, 172.3.


**3-((2-(Acetoxy)ethoxy)methyl)-6-(4-*tert*-butylphenyl)-3H-furano [2,3-d]-pyrimidine-2-one (20b)**


Compound **20b** was obtained as described for **20a** using 1-ethynyl-4-*tert*-butylbenzene (0.72 mmol, 114 mg). The yield of **20b** as a white powder was 98 mg (53%). ^1^H-NMR (CDCl_3_) δ, ppm: 1.37 (9H, s, (CH_3_)_3_), 2.08 (3H, s, CH_3_), 3.87–3.98 (2H, m, CH_2_), 4.20–4.28 (2H, m, CH_2_), 5.52 (2H, s, CH_2_), 6.69 (1H, s, H-5), 7.46–7.53 (2H, m, Ph), 7.72 (1H, d, *J* = 8.4 Hz, Ph), 8.03 (1H, s, H-4). ^13^C NMR (CDCl_3_) δ, ppm: 20.8, 29.7, 31.1, 34.9, 63.1, 68.5, 76.8, 79.5, 96.4, 109.3, 125.0, 126.0, 137.8, 143.0, 153.6, 155.7, 156.8, 170.7, 172.4.


**3-((2-(Hydroxy)ethoxy)methyl)-6-(4-pentylphenyl)-3H-furano [2,3-d]-pyrimidine-2-on (5) and 3-((2-(Hydroxy)ethoxy)methyl)-6-(4-pentylphenyl)-3H-pyrrolo [2,3-d]-pyrimidine-2-one (7)**


Compound **20a** (0.25 mmol, 100 mg) was dissolved in 32% NH_3_ in methanol (15 mL). The reaction mixture was kept at 36 °C for 72 h. The solvent then was evaporated and a new portion of 32% NH_3_ in methanol was added (15 mL). The procedure was repeated until it reached approximately 50% conversion of compound **5** to compound **7**, controlling the progress of the reaction by TLC. The solvent then was evaporated and the products were purified using preparative chromatography in CHCl_3_:CH_3_OH (9:1) system.

The yield of **5** as a pale yellow powder was 31 mg (35%). ^1^H-NMR (CD_3_OD) δ, ppm: 0.82–0.89 (3H, m, CH_3_), 1.23–1.29 (4H, m, 2xCH_2_), 1.58–1.63 (2H, m, CH_2_), 2.59–2.64 (2H, m, CH_2_), 3.71 (2H, s, CH_2_), 3.83–3.87 (2H, m, CH_2_), 5.47 (2H, s, CH_2_), 6.51 (1H, s, H-5), 7.23–7.25 (2H, m, Ph), 7.58–7.61 (2H, m, Ph), 8.27 (1H, s, H-4). ^13^C NMR (CD_3_OD) δ, ppm: 13.8, 22.4, 29.6, 30.9, 31.4, 35.6, 61.0, 71.5, 80.2, 96.1, 111.8, 125.3, 129.1, 139.7, 142.2, 144.6, 154.3, 157.9. HRMS m/z: calculated for C_20_H_24_N_2_O_4_ [M+H]+ 357.1809; found [M+H]^+^ 357.1817.

The yield of **7** as a pale yellow powder was 36 mg (40%). ^1^H-NMR (CD_3_OD) δ, ppm: 0.86–0.90 (3H, m, CH_3_), 1.29–1.36 (4H, m, 2xCH_2_), 1.58–1.64 (2H, m, CH_2_), 2.60–2.65 (2H, m, CH_2_), 3.71 (2H, s, CH_2_), 4.00–4.05 (2H, m, CH_2_), 5.47 (2H, s, CH_2_), 6.75 (1H, s, H-5), 7.23–7.26 (2H, m, Ph), 7.64–7.68 (2H, m, Ph), 8.27 (1H, s, H-4). ^13^C NMR (CD_3_OD) δ, ppm: 13.8, 22.4, 30.8, 31.3, 35.7, 55.6, 60.9, 71.7, 80.1, 96.7, 109.7, 125.0, 125.4, 129.1, 139.2, 145.5, 156.9, 159.0. HRMS m/z: calculated for C_20_H_25_N_3_O_3_ [M+H]+ 356.1977; found [M+H]^+^ 356.1969.


**3-((2-(Hydroxy)ethoxy)methyl)-6-(4-tert-butylphenyl)-3H-pyrrolo [2,3-d]-pyrimidine-2-one (14)**


Compound **20b** (0.28 mmol, 100 mg) was dissolved in 32% NH_3_ in methanol (15 mL). The reaction mixture was kept at 36 °C for 72 h. The solvent then was evaporated and a new portion of 32% NH_3_ in methanol was added (15 mL). The progress of the reaction was controlled by TLC. The solvent then was evaporated and the product was purified using preparative chromatography in the CHCl_3_:CH_3_OH (9:1) system. The yield of **14** as a yellow powder was 34 mg (54 %). ^1^H NMR (CD_3_OD) δ, ppm: 1.37 (9H, s, (CH_3_)_3_), 3.80–3.69 (4H, m, 2xCH_2_), 5.55 (2H, s, CH_2_), 6.75 (1H, s, H-5), 7.56–7.54 (2H, m, Ph), 7.74–7.66 (2H, m, Ph), 8.59 (1H, s, H-4). ^13^C NMR (CD_3_OD) δ, ppm: 30.2, 34.3, 60.6, 71.4, 80.2, 96.8, 125.0, 125.5, 125.7, 126.9, 127.2, 141.5, 152.4, 154.1, 157.4. HRMS m/z: calculated for C_20_H_25_N_3_O_3_ [M+H]+ 342.1812; found [M+H]^+^ 342.1814.

### 4.2. Virology Assays

#### 4.2.1. General

Vero E6 cells from African green monkey (*Chlorocebus aethiops*) kidney and Madin-Darby canine kidney (MDCK) cells were from the Russian National Collection of Cell Cultures at the N.F. Gamaleya National Research Center for Epidemiology and Microbiology, of the Ministry of Health of the Russian Federation (Moscow, Russia). The cells were cultivated in DMEM supplemented with 2.5% FetalClone II (Thermo Fischer Scientific, Waltham, MA, USA) (Vero E6 cells) or in Eagle’s minimal essential medium supplemented with 5% fetal bovine serum (FBS) (HyClone, Waltham, MA, USA) 2xnon-essential amino acids (Gibco), 10 mM glutamine, and 4% gentamycin at 37 °C in a humid atmosphere containing 5% CO_2_. Every 2–3 weeks they were checked for mycoplasma contamination by standard PCR.

Stock solutions of all tested compounds were prepared in DMSO which were diluted in the respective media before each experiment.

Influenza A/California/7/2009 (H1N1)pdm09 strain and human coronavirus SARS-CoV-2 (passage 4) corresponding to hCoV-19/Russia/Moscow-PMVL-12/2020 strain with infectivities of 10^7.25^ TCID_50_/_mL_ and 10^6^ TCID_50_/_mL_, respectively, were from the Russian State collection of viruses at the National Research Center for Epidemiology and Microbiology. Influenza virus was cultivated in the allantoic cavities of 9–10-day-old chicken eggs for 48 h at 36°C. Infectious and hemagglutination activities were quantified according to the protocols recommended by the World Health Organization (WHO) [40]. The multiplicity of infection for SARS-CoV-2 was determined by immunostaining of Vero E cells as described in [41].

#### 4.2.2. SARS-CoV-2 Assays

Vero E6 cells were seeded 24 h prior to infection in 24-well plates at 5 × 10^4^ cells/well in DMEM supplemented with 2.5% FBS. When the cells reached >95% confluency, they were inoculated with pre-diluted SARS-CoV-2 at MOI of 0.1. Two hours later the medium was removed, the cells were washed, and fresh medium containing the tested compounds was added. Total RNA was purified from the conditioned medium 4 days post-infection using the High Pure RNA Isolation Kit (Roche Life Sciences, Switzerland) according to the manufacturer’s instructions. Each compound was tested in triplicate. Reverse transcription and quantification of SARS-CoV-2 genomic RNA was performed as described in [42]. The antiviral activity was expressed as mean logarithmic changes in the virus RNA level in comparison to DMSO-treated cells (Δlog_10_(RNA)).

Alternatively, Vero E6 cells were seeded on 96-well plates at a density of 1.2 × 10^4^ cells/well, infected with serial dilutions of SARS-CoV-2 and incubated with compounds as described above, and the virus titer was assessed by determination of the endpoint dilution at which it exhibits 50% of its maximal cytopathic effect (TCID_50_). The antiviral activity was expressed as a logarithmic change in the TCID_50_ value in the presence of a compound compared with the DMSO-treated control (Δlog_10_(TCID_50_)). Each compound was tested in quadruplicate.

#### 4.2.3. Influenza Virus Assay

MDCK cells were seeded onto 96-well plates in Eagle’s minimal essential medium supplemented with 2xnon-essential amino acids (Gibco), 0.2% bovine serum albumin, and 2 µg/mL TPCK-treated trypsin (Sigma, Darmstadt, Germany). All compounds were added 1 h prior to the infection in 100 µL of medium. Then 10-fold serial dilutions of influenza virus were added. Changes in the virus titers in the presence of 50 μM compounds were quantified by measuring changes in the TCID_50_ values (Δlog_10_(TCID_50_)) as described above. To determine the half-inhibitory concentrations (IC_50_), the TCID_50_ values were measured for serial dilutions of each compound, and concentrations at which TCID_50_ was reduced by 50% were calculated. The cytotoxic dose (CD_50_) value was determined as the concentration of each compound at which it decreased cell number by 50%. The selectivity index was calculated as a ratio of CD_50_ to IC_50_ values. Each compound was tested in three independent experiments with technical duplicates in each of them.

### 4.3. Biochemical Assays

#### 4.3.1. RdRp and RNA Substrates

SARS-CoV-2 RdRp holoenzyme containing the catalytic subunit nsp12 and fused nsp7-nsp8 subunits (separated by a His6-tag) was expressed in *E. coli* BL-21(DE3) and purified using Ni-affinity and anion-exchange chromatography steps as described previously [32,43]. RNA oligonucleotides for in vitro reactions were prepared by DNA synthesis (Moscow, Russia). All reagents were from Sigma-Aldrich (St. Louis, USA) unless otherwise specified. The inhibitors were diluted in DMSO (VWR International LLC, USA).

#### 4.3.2. In Vitro Transcription Assay

The primer RNA oligonucleotide (Figure 3) was 5′-labeled with γ-[^32^P]ATP and T4 polynucleotide kinase (NEB, Ipswich, USA). The labeled primer was mixed with the template RNA oligonucleotide at a molar ratio of 1:1.2 in the reaction buffer containing 10 mM Tris-HCl, pH 7.9, 10 mM KCl, 2 mM MgCl_2_, and 1 mM DTT. The samples were incubated for 3 min at 95 °C, cooled down to 85 °C for 2 min, and then to 25 °C for 2 h. The annealed RNA duplex was stored at -20 °C. To test the effects of various compounds on the RdRp activity, reactions in vitro were performed with different orders of the addition of the reaction components. In the first case, RdRp (1 μM final concentration) was incubated with the inhibitors (1 mM, or the same volume of DMSO in control samples) for 5 min at 30 °C, then the RNA substrate (25 nM) was added, and the samples were incubated for 10 min at 30 °C. In the second case, RdRp was mixed with the RNA substrate and the inhibitors and incubated for 10 min at 30 °C. In the third case, the RNA substrate was pre-incubated with inhibitors for 5 min at 30 °C, then RdRp was added, and the samples were incubated for 10 min at 30 °C. In all cases, RNA synthesis was initiated by adding 100 μM ATP (Thermo Fisher Scientific, Waltham, USA) and was quenched after 1 min with an equal volume of the stop solution containing 8 M urea, 20 mM EDTA, and 2×TBE. The reaction products were separated by 18% denaturing PAGE (acrylamide:bisacrylaimde 19:1) and visualized by phosphorimaging using a Typhoon 9500 scanner (GE Healthcare). The efficiency of RNA extension was calculated in every sample and normalized to the activity in the absence of inhibitors. To measure the IC_50_ values, RdRp was incubated with increasing concentrations of inhibitors (1 μM, 3 μM, 10 μM, 30 μM, 100 μM, 300 μM, 1000 μM, or 2500 μM), then the RNA substrate and ATP were added, and the reactions were performed as described above. The data were fitted to the hyperbolic equation A = Amax × (1 – [Inhibitor]/(IC_50_ + [Inhibitor])), where A is the efficiency of the RNA extension at a given concentration of the inhibitor and Amax is the maximal activity.

#### 4.3.3. Electrophoretic Mobility Shift Assay

Reaction mixtures containing RdRp (1 μM final concentration) and the RNA substrate (25 nM) were assembled in the reaction buffer containing BSA (100 μg/mL) as described above. The inhibitors (1 mM) were added either before or after the RNA substrate. The samples were incubated for 10 min at 30 °C, mixed with 5x loading buffer (2.5^×^ TBE, 50% glycerol, 0.05% bromophenol blue) and separated by 5% nondenaturing PAGE (acrylamide:bisacrylaimde 37.5:1, 0.5^×^ TBE, 10 V × cm^−1^) at 4 °C. The gels were analyzed by phosphorimaging and the percentage of the binary complex was calculated for every sample.

#### 4.3.4. Molecular Docking

The structure of SARS-CoV-2 nsp12 (PDB: 6M71) [34] was used as the target for molecular docking after preparation in the program UCSF Chimera [44] as described previously [45]. The 3D structure of compound 1 was created with the Open Babel chemical toolbox [46] and used as the ligand. Both the target and the ligand were uploaded to the SwissDock web service [35]. The simulation was carried out with default standard parameters. The result was visualized using UCSF Chimera.

## Data Availability

The data presented in this study are available within the article.

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
