# Peer review of "Nucleoside Analogs That Inhibit SARS-CoV-2 Replication by Blocking Interaction of Virus Polymerase with RNA"

_ijms, 2023, doi:10.3390/ijms24043361_

Round 1

Reviewer 1 Report

This manuscript described several various analogues of the leader compounds, the evaluation of the cytotoxicity and antiviral activity of the obtained compounds against SARS-CoV-2. Although the author did a lot of work, but current version is not convincing. The structural modification didn’t increase activity of the compounds and some conclusion is still need be confirmed. Reviewer doesn’t recommend this manuscript be published in IJMS for the following reasons:

1.        As to antiviral compounds, 50% effective concentration (EC50) must be given. Comparison of EC50s between parent compounds 1&2, NHC and the analogs were missing.  

2.        The results of the primer extension and EMSA data were inconsistent with the data in table 1. If the working machoism was as author claimed, the compound binding to the RdRp then inhibitor the RNA processing. These two sets of data should be agreed to each other.

 3.        Docking study of the compounds to the RdRp is necessary.

 4.        Some mislabeling: ie compound 12 and 13, no R group!

Author Response

The authors are very grateful to the Reviewer for the valuable comments, which allowed us to improve the quality of the article. All comments have been addressed, the article has been corrected accordingly.

This manuscript described several various analogues of the leader compounds, the evaluation of the cytotoxicity and antiviral activity of the obtained compounds against SARS-CoV-2. Although the author did a lot of work, but current version is not convincing. The structural modification didn’t increase activity of the compounds and some conclusion is still need be confirmed. Reviewer doesn’t recommend this manuscript be published in IJMS for the following reasons:

  1. As to antiviral compounds, 50% effective concentration (EC50) must be given. Comparison of EC50s between parent compounds 1&2, NHC and the analogs were missing.  

We thank the reviewer for this comment. We usually measure changes in the virus RNA content and infectivity at a fixed concentration of compounds for initial screening, giving Δlog10 values presented in the manuscript. If a compound decreases the virus RNA level by at least 3 logs at concentration of 50 uM, the next step is to determine EC50 by testing different doses of the compound. In this study, the activities of all tested derivatives of the leader compounds were moderate or low, with the maximal changes in the RNA level or virus infectivity within 1.5 orders of magnitude, which means that EC50 values would be within 20-50 uM concentrations. In fact, the Δlog10 values presented in the manuscript allow comparison of the antiviral activity of compounds within the dynamic range of 4-6 logs, as shown for the NHC control, which changes the RNA level by 4 orders of magnitude. In the revised manuscript, we have discussed these results in more detail and compared the antiviral activity of various compounds:

“The antiviral activity of all synthesized compounds towards SARS-CoV-2 was assessed by two approaches. First, we quantified changes in the amount of viral RNA during virus replication in the conditioned medium (to measure the reduction of virion production) in the presence of each compound taken at a fixed concentration (Dlog10(RNA)) (Table 1). Second, we analyzed changes in the infectivity of the virus, by measuring changes in the Tissue Culture Infectious Dose values in Vero E6 cells (Dlog10(TCID50)). A previously studied anti-SARS-CoV-2 agent N4-hydroxycitidine (NHC) was used as a control [26,29].

These two approaches gave complementary results, presented in Table 1. Two of the compounds (6 and 7) were inactive towards SARS-CoV-2 in both assays. The highest activity was displayed by the previously described compound 1, which was active in both assays and significantly decreased the amount of viral RNA (>10-fold) and the TCID50 value (~10-fold). Moderate activity was registered for the newly developed compounds 3, 5, 8, 11, 13 and 14, which could inhibit virus replication or infectivity at least 10-fold in one of the two assays (Table 1). Compound 8, albeit showing visible reduction in the viral RNA titer, provided weak protection of Vero cells in the infectivity test. On the contrary, compounds 5 and 14 only weakly affected the RNA titer but had stronger effects on the TCID50 values. Other tested compounds showed weak antiviral activity in both assays. The toxicity of all tested compounds was measured by the standard MTT assay in Vero E6 cells (CD50 Vero E6, Table 1). Noteworthy, all of them were almost nontoxic at the effective concentration, as just few of them inhibited cell growth at concentrations of 150 μM or above.”

  1. The results of the primer extension and EMSA data were inconsistent with the data in table 1. If the working machoism was as author claimed, the compound binding to the RdRp then inhibitor the RNA processing. These two sets of data should be agreed to each other.

We are grateful to the reviewer for these comments. In general, the data on infectious activity are consistent with the inhibitory activity of the tested compounds in the primer extension assay. It should be noted that not all compounds analyzed in cell-based assays were tested with RdRp in vitro. For this analysis, we selected 6 compounds that had the strongest effects in cell cultures and 3 compounds that only weak effects. Notably, the efficiency of RdRp inhibition by most compounds from the first group was higher obtained than that for compounds from the second group (Fig. 3). Furthermore, the IC50 values for RdRp inhibition by the three selected compounds (1, 5 and 14) measured in vitro (Fig. 5) were in the same range of concentrations that were active in cell-based assays (around 50 uM). Some discrepancies between the efficiencies of inhibition of RdRp activity in vitro and the virus replication in vivo could be likely explained by different cell permeability and metabolism of various compounds. We have discussed these issues in the revised manuscript:

Results:

“The reactions were performed with compounds 1, 3, 5, 8, 11 and 14, which noticeably inhibited virus replication in cell-based assays, and with compounds 9, 10 and 12, which showed weaker effects in cell cultures. The strongest effects were observed for compounds 1, 3, 5, 10 and 14, which completely blocked RNA synthesis. Overall, these data corroborate our findings from the cell infectivity models. In particular, four of these compounds (1, 3, 5 and 14) are also among the strongest inhibitors in cell-based assays. Compound 10, while being less efficient in cell culture, could nevertheless fully inhibit RdR in vitro. Furthermore, compounds 9 and 12, which are less efficient in cell-based assays, also have weaker effects on the RdRp activity in vitro (Figure 3A, 3D).”

Discussion:

“In particular, compounds 1, 3, 5 and 14 can inhibit both virus replication in cell culture and RdRp activity in vitro, while several compounds that are less efficient in cell culture also have lower activity against RdRp in vitro. The IC50 values for inhibition of RdRp activity by compounds 1 and 5 (Figure 5) exactly correspond to the concentration at which they can inhibit virus replication in cell-based assays (Table 1). Although the IC50 value for compound 14 is somewhat higher than its active concentration in cell culture, it can also fully inhibit RdRp in vitro. This suggests that RdRp is likely the natural target of these compounds during virus replication. Some discrepancies between the inhibitory activities of the tested compounds in RdRp assays and in cell-based assays (e.g., relatively low activity of compound 10 in cell cultures despite its ability to fully inhibit RdRp in vitro) can likely be explained by different cell permeability and/or differences in the metabolism of these compounds in vivo.”

  1. Docking study of the compounds to the RdRp is necessary.

We thank the reviewer for this suggestion. We have performed docking of the lead compound 1, which was active in both cell-based assays and in reactions with RdRp in vivo, on the RdRp structure using the SwissDock web portal. Consistently with the relatively weak activity of the tested compounds, this docking did not reveal a particular binding site with the highest binding energy but rather identified multiple binding clusters with comparable energies (from -6 to -7.5 kcal/mol). Comparison of these binding sites with available structures of SARS-CoV2 RdRp with RNA substrates reveals that several of the predicted sites are located within the RNA binding channel of RdRp, in agreement with the proposed mechanism of action of these compounds identified in biochemical experiments. Further experiments are required to establish  whether these compounds target a single site or may bind to several alternative sites on RdRp. However, we think that these experiments should be performed after finding better versions of the inhibitors, which would have higher affinity to RdRp and higher efficiency in vivo. These observations are presented in Fig. 6 and are described in Discussion:

“Since the tested compounds can inhibit RdRp activity by preventing its interactions with RNA, we propose that their binding sites are likely located within the RNA binding channel of RdRp. Docking of compound 1 on the SARS-CoV-2 RdRp structure using the SwissDock web service revealed multiple potential binding sites, some of which were indeed located within the RNA-binding channel (Figure 6). However, no preferred site with the highest binding energy could be revealed in this modeling, consistent with the relatively low affinity (IC50 values) of the tested compounds to RdRp. Thus, nucleoside derivatives tested here might potentially target several alternative sites on the RNA binding surface of RdRp and further experiments are required to establish their exact binding mode to the replication complex of SARS-CoV-2.”

  1. Some mislabeling: ie compound 12 and 13, no R group!

Thank you. Fixed.

Reviewer 2 Report

The manuscript of Matyugina et al. “Nucleoside analogs that inhibit SARS-CoV-2 replication by blocking interaction of virus polymerase with RNA” describes the synthesis of novel nucleoside analogs and assessment of their anti-viral activity against SARS-CoV-2 and influenza virus as well as determination of their mode of action. The results suggest that novel compounds block the interaction of viral RNA polymerase wirg RNA instead of incorporation into nascent RNA chain.

Prior to publishing, several issues in the manuscript should be addressed.

Table 1. Please provide the data of anti-viral activity of reference compound for influenza virus. Also, the standard deviation values should be given for CC50’s and IC50’s.

Please provide the reference confirming the statement “…a well-known anti-SARS-CoV-2 agent N4-hydroxycitidine (NHC)…”

The results of anti-viral assays for SARS-CoV-2 and influenza virus are given in different units, Δ lgmax for SARS-CoV-2 and selectivity index for influenza virus. No problems are with influenza virus, as the higher SI, the more prospective the compound. Please discuss the potency of the compounds against SARS-CoV-2. If, for instance, Δ lgmax is 1.0 – does this mean the compound is strong or not?

What was the dose of viruses used for the CPE reduction assay?

Page 7 section 3. Change to read “can also inhibit the activity of the SARS-CoV-2 RdRp in in vitro assays”.

Author Response

The authors are very grateful to the Reviewer for the positive assessment of our work and valuable comments, which allowed us to improve the quality of the article. All comments have been addressed; the article has been corrected accordingly.

The manuscript of Matyugina et al. “Nucleoside analogs that inhibit SARS-CoV-2 replication by blocking interaction of virus polymerase with RNA” describes the synthesis of novel nucleoside analogs and assessment of their anti-viral activity against SARS-CoV-2 and influenza virus as well as determination of their mode of action. The results suggest that novel compounds block the interaction of viral RNA polymerase wirg RNA instead of incorporation into nascent RNA chain.

Prior to publishing, several issues in the manuscript should be addressed.

Table 1. Please provide the data of anti-viral activity of reference compound for influenza virus. Also, the standard deviation values should be given for CC50’s and IC50’s.

Oseltamivir was used as a reference compound for influenza virus (IC50 = 0.61 uМ, СD50 =500 uМ and SI = 820). The standard deviation (SD) values have been added to the Table.

Please provide the reference confirming the statement “…a well-known anti-SARS-CoV-2 agent N4-hydroxycitidine (NHC)…”

Done

The results of anti-viral assays for SARS-CoV-2 and influenza virus are given in different units, Δ lgmax for SARS-CoV-2 and selectivity index for influenza virus. No problems are with influenza virus, as the higher SI, the more prospective the compound. Please discuss the potency of the compounds against SARS-CoV-2. If, for instance, Δ lgmax is 1.0 – does this mean the compound is strong or not?

Thank tyou for these comments. In the revised manuscript, we have described the cell-based assays in more detail and explained the meaning of the delta-log parameters. The Δlog10(RNA) values (previously designated Δlgmax) represent changes in the virus RNA titer, and Δlog10(TCID50) show changes in the virus infectivity at the tested concentration of compounds. The higher value of Δlog10 indicates stronger antiviral activity. According to its definition, Δlog10 equal to 1.0 corresponds to a ten-fold change in the RNA level or virus infectivity. In our case, the highest activity towards SARS-CoV-2 was shown by compound 1. The activity of the most compounds was moderate or low, as their Δlog10 values were still lower than in the case of NHC that is used for treatment of COVID-19 patients.

We have changed the description of these experiments in the Results section as follows:

“The antiviral activity of all synthesized compounds towards SARS-CoV-2 was assessed by two approaches. First, we quantified changes in the amount of viral RNA during virus replication in the conditioned medium (to measure the reduction of virion production) in the presence of each compound taken at a fixed concentration (Dlog10(RNA)) (Table 1). Second, we analyzed changes in the infectivity of the virus, by measuring changes in the Tissue Culture Infectious Dose values in Vero E6 cells (Dlog10(TCID50)). A previously studied anti-SARS-CoV-2 agent N4-hydroxycitidine (NHC) was used as a control [26,29].

These two approaches gave complementary results, presented in Table 1. Two of the compounds (6 and 7) were inactive towards SARS-CoV-2 in both assays. The highest activity was displayed by the previously described compound 1, which was active in both assays and significantly decreased the amount of viral RNA (>10-fold) and the TCID50 value (~10-fold). Moderate activity was registered for the newly developed compounds 3, 5, 8, 11, 13 and 14, which could inhibit virus replication or infectivity at least 10-fold in one of the two assays (Table 1). Compound 8, albeit showing visible reduction in the viral RNA titer, provided weak protection of Vero cells in the infectivity test. On the contrary, compounds 5 and 14 only weakly affected the RNA titer but had stronger effects on the TCID50 values. Other tested compounds showed weak antiviral activity in both assays. The toxicity of all tested compounds was measured by the standard MTT assay in Vero E6 cells (CD50 Vero E6, Table 1). Noteworthy, all of them were almost nontoxic at the effective concentration, as just few of them inhibited cell growth at concentrations of 150 μM or above.”

What was the dose of viruses used for the CPE reduction assay?

The infectivity of influenza virus and SARS-CoV-2 stock were 107.25 TCID50/ml and 106 TCID50/ml, respectively. In the CPE reduction assay these viruses were titrated starting with a 10-fold dilution, with the highest doses being 106.25 TCID50/ml and 105 TCID50/ml.

Page 7 section 3. Change to read “can also inhibit the activity of the SARS-CoV-2 RdRp in in vitro assays”.

Done

Round 2

Reviewer 1 Report

I would like to suggest to accept manuscript in current version.